# Poorer verbal working memory for a second language selectively impacts academic achievement in university medical students

Collette Mann, Benedict J. Canny, David H. Reser and Ramesh Rajan

Department of Physiology, Faculty of Medicine Nursing and Health Sciences, Monash University, Clayton, VIC, Australia

Corresponding author
Collette Mann,
Collette.Mann@med.monash.edu.au

## ABSTRACT

Working memory (WM) is often poorer for a second language (L2). In low noise conditions, people listening to a language other than their first language (L1) may have similar auditory perception skills for that L2 as native listeners, but do worse in high noise conditions, and this has been attributed to the poorer WM for L2. Given that WM is critical for academic success in children and young adults, these speech in noise effects have implications for academic performance where the language of instruction is L2 for a student. We used a well-established Speech-in-Noise task as a verbal WM (vWM) test, and developed a model correlating vWM and measures of English proficiency and/or usage to scholastic outcomes in a multi-faceted assessment medical education program. Significant differences in Speech-Noise Ratio ($SNR_{50}$) values were observed between medical undergraduates who had learned English before or after five years of age, with the latter group doing worse in the ability to extract whole connected speech in the presence of background multi-talker babble (Student-t tests, $p < 0.001$). Significant negative correlations were observed between the $SNR_{50}$ and seven of the nine variables of English usage, learning styles, stress, and musical abilities in a questionnaire administered to the students previously. The remaining two variables, Perceived Stress Scale (PSS) and the Age of Acquisition of English (AoAoE) were significantly positively correlated with the $SNR_{50}$, showing that those with a poorer capacity to discriminate simple English sentences from noise had learnt English later in life and had higher levels of stress – all characteristics of the international students. Local students exhibited significantly lower $SNR_{50}$ scores and were significantly younger when they first learnt English. No significant correlation was detected between the $SNR_{50}$ and the students' Visual/Verbal Learning Style ($r = -0.023$). Standard multiple regression was carried out to assess the relationship between language proficiency and verbal working memory ($SNR_{50}$) using 5 variables of L2 proficiency, with the results showing that the variance in $SNR_{50}$ was significantly predicted by this model ($r^2 = 0.335$). Hierarchical multiple regression was then used to test the ability of three independent variable measures ($SNR_{50}$, age of acquisition of English and English proficiency) to predict academic performance as the dependent variable in a factor analysis model which predicted significant performance differences in an assessment requiring communications skills ($p = 0.008$), but

not on a companion assessment requiring knowledge of procedural skills, or other assessments requiring factual knowledge. Thus, impaired vWM for an L2 appears to affect specific communications-based assessments in university medical students.

## INTRODUCTION

In medical education, most information is communicated verbally, often to large groups of students. Consequently, listening abilities and language comprehension are critical to learning and require both auditory perception and auditory working memory (WM) skills. WM is defined as *"the system for the temporary maintenance and manipulation of information, necessary for the performance of such complex cognitive activities as comprehension, learning, and reasoning..."* (*Baddeley, 1992*, p. 281). One core element of WM, and in particular verbal Working Memory (vWM), is the "phonological loop", which has been shown to be critical for language acquisition during development, as well as language processing in daily life (*Baddeley, 1992*). However, it has been widely reported that WM capacity may be limited for students who are learning in an environment where the language of instruction is not their native language (*Andersson, 2010*; *Kroll et al., 2002*; *Mackey et al., 2002*; *McDonald, 2006*; *Miyake & Friedman, 1998*; *Service, 1992*; *Service et al., 2002*; *Sunderman & Kroll, 2009*; *Tokowicz, Michael & Kroll, 2004*) and this appears to be due to demands on WM resources in the second language (L2) (*Service et al., 2002*).

The relationship between WM capacity and academic achievement has been well studied in children (*Alloway & Elsworth, 2012*; *Gathercole & Pickering, 2000a*; *Gathercole & Pickering, 2000b*; *Gathercole et al., 2004*; *Vock & Holling, 2008*) and in university students and adults (*Daneman & Carpenter, 1980*; *Daneman & Hannon, 2001*; *Swanson, 1994*; *Tolar, Lederberg & Fletcher, 2009*). Whilst the studies in younger learners have shown strong correlations between WM and high academic attainment (*Alloway & Alloway, 2010*; *Gathercole et al., 2004*; *St Clair-Thompson & Gathercole, 2006*), studies of university science students have reported that WM has only weak or indirect effects in predicting academic performance (*Krumm, Ziegler & Buehner, 2008*; *Rohde & Thompson, 2007*). *Tolar, Lederberg & Fletcher (2009)* found WM strongly related to the adults' ability on Scholastic Aptitude Test (SAT) scores, but effects were reduced when other cognitive factors were controlled for, such as spatial ability. Further, some studies suggest that vWM may not have as great an effect on the students' processing abilities as the direct effects of the students' first language (L1), including the ability to suppress L1 influences or the level of L1 proficiency and general language aptitude (for review see *Juffs & Harrington, 2011*).

In addition or in consequence of the poorer vWM for L2, the acoustic environment to facilitate ideal listening conditions may also be crucial for effective learning by L2

medical undergraduates. It has been noted that non-native listeners may have similar speech perception skills as native listeners in low noise conditions, but that these abilities significantly decrease in high noise conditions (*Buus et al., 1986*; *Florentine et al., 1984*; *Lin, Chang & Cheung, 2004*; *Mayo, Florentine & Buus, 1997*; *Tabri, Abou Chacra & Pring, 2011*; *Takata & Nabelek, 1990*). Using the Speech-in-Noise (SiN) task, *Mayo, Florentine & Buus (1997)* showed that not only was speech perception in noise poorer in L2 learners, but that it was also dependent on the age the L2 was acquired; bilinguals who learnt English after 14 years of age had the worst performance in the SiN task compared to monolinguals and bilinguals who learnt English before 6 years of age. Further, in contrast to the monolinguals, the late bilinguals did not benefit from contextual cues in those sentences that were highly predictive (i.e. sentences in which the subjects could easily guess the target word). Similarly, *Buus et al. (1986)* found that the noise tolerance level of non-native listeners to understand 50% of the test sentences, increased with years of exposure to English, but never reached the level of tolerance (and achievement) of a native English speaker.

There is evidence that the ability to process speech in noise influences the ability to recall academic material. *Ljung et al. (2010)* tested 48 native Swedish university students with open-ended questions about the content of spoken lectures of up to eight minutes duration presented in broadband noise or quiet, or presented students with 10 paragraphs of lectures in classrooms of differing reverberation times. The subjects' memory performance was significantly worse under both adverse conditions compared with the quiet condition, even when the students had heard correctly the spoken lectures.

Given the relationship between vWM capacity, academic achievement and the impairment of speech comprehension in noisy environments by L2 learners, such effects are likely to be even stronger for these students. Thus, a potential disadvantage exists for medical students learning a course in their L2. This is particularly relevant to the many international medical students that travel to mainly English-speaking western universities in, e.g., Australia, the UK or the USA (*Brisset et al., 2012*) especially those for whom the L2 was not acquired at an early age. Our study has important implications in identifying another significant factor impacting on the academic performance in the early years of a medical undergraduate course, the period of greatest stress and of greatest likelihood of drop-outs/failures (*Baker, 2004*).

In the present study we examined the relationships between vWM for L2, the age at which the L2 was acquired, and students' scholastic outcomes. In a previous study (*Mann et al., 2010*), we showed that international students in a Bachelor of Medicine/Bachelor of Surgery (MBBS) course in an Australian university performed worse than their local peers, but that this was significantly influenced by the students' L1. This is consistent with the idea that L1 influences may affect academic outcomes for instruction in an L2. Building on this, we now explore whether verbal WM plays a role in the academic achievements of a cohort of international and local medical undergraduates in the same course. Specifically, we hypothesise that 1) students with English as a Second Language (ESL students) will have lower scores than students with English as a First Language (EFL students) in the SiN test

(reflecting poorer vWM); and 2) that the students with lower SiN results will also have lower academic scores in their different assessments.

As well as having a high secondary school result (a pre-requisite also for local students), international medical students must pass stringent measures of English proficiency prior to enrolment and must also attend and pass an interview to demonstrate high motivation and self-expectations. To a major extent these requirements obviate the confounding effects of English proficiency skills often suggested (*Lun, Fischer & Ward, 2010*; *Webb, 2002*) to account for the fact that, generally, international medical students do not perform as well academically as their local counterparts (*Bagot et al., 2005*; *Liddell & Koritsas, 2004*; *Wass et al., 2003*). We used a well-established auditory test paradigm as a vWM test, free of L2 proficiency concerns that have been raised against such tests as the Reading Span Test (RST) when applied to L2 learners (*Juffs & Harrington, 2011*). The SiN task tests vWM via the phonological loop through storing, processing and recall of speech in background noise.

## MATERIAL AND METHODS

### Participants

All participants in this study were students enrolled in the MBBS program from 2008 to 2010 at Monash University. The students were informed that this project was biphasic and participation involved both completing a questionnaire and an invitation at a later date to undergo an audiometry test. The questionnaire asked for information on the students' personal demographics, English acquisition and usage, musical abilities and two psychometric measures: Perceived Stress Scale (*Cohen, 1994*) and the Index of Learning Styles Questionnaire (*Felder & Soloman, 1994*). Stress has been found to have a negative impact on the academic performance of first year medical students, particularly international students (*Bagot et al., 2005*; *Baker, 2004*; *Lacina, 2002*; *Mori, 2000*) as well as the style of learning adopted by international versus local students, such as deep vs. surface learning styles (*Bagot et al., 2005*; *Newble & Entwistle, 1986*; *Volet, Renshaw & Tietzel, 1994*; *Zeegers, 2001*). As mentioned in the Introduction, the international medical students of this course must pass stringent measures of English proficiency prior to enrolment, such as the International English Language Testing System (IELTS) or the Test Of English as a Foreign Language (TOEFL). Therefore, the questions on the survey pertained mainly to measurable English attributes such as 'In what order did you learn English and your other language'? There was one question on the students' perceived English and Language Other Than English (LOTE) proficiency which was purely self-rated from a score of '0 = poor' to '4 = excellent'.

The surveys were distributed at the commencement of each university year in the 1st year of the medical undergraduates' course. Of the 791 questionnaires distributed over the three years, 582 were returned giving a response rate of 73.6%. Participation was voluntary and students could withdraw at any stage.

In the second phase of the project, students were asked to participate in a SiN test (described below). As it was not feasible to submit all 582 subjects to this test, we

performed a power analysis using GPower 3.0.10, which calculated that we would require 15 subjects in each group to give us an effect size of 0.8 at a power level of 90%. We then emailed all 582 students inviting them to attend the audiometry test at a mutually convenient time. From these emails, we had a total of 113 subjects that came in to be tested on the speech-in-noise task. Of these 113, ten participants were excluded from data analysis: one subject was excluded due to hearing impairments and nine candidates were classed as outliers with means more than two standard deviations from the sample mean (at $\alpha = 0.05$), leaving a total of 103 subjects tested and analysed, which still gave us ample power for this particular study. Analysis and findings relevant to all 582 students (including the 113 who participated in the audiometry tests) are currently being researched by the authors, and will be reported elsewhere; the emphasis of this report is on the outcomes of the 103 subjects undertaking the SiN test.

Demographic characteristics are set out in Table 1.

Students were classed as 'local' if they were Australian or New Zealand citizens, or if they held permanent residency for more than three years; or students were classed as 'international' if they held temporary entry visas, in accordance with the option chosen by the students on their questionnaires. Only one student held permanent residency status and had been living in Australia for over five years; all other students were citizens or held temporary entry visas.

## Audiometry testing

At the outset, hearing sensitivity in each subject was measured with audiometry using a Beltone Model 110 Clinical Audiometer, calibrated to present pure tones through calibrated TDH headphones. Hearing was tested one ear at a time at 500 Hz, 1000 Hz, 2000 Hz, 4000 Hz, 6000 Hz and 8000 Hz. The minimum sound level at each frequency was recorded as the threshold in decibels Hearing Level (dB HL) relative to normal hearing sensitivity (*International Organization for Standardization, 1989*). We then calculated the bilateral four tone threshold average from thresholds at 500 Hz, 1000 Hz, 2000 Hz and 4000 Hz. Generally, only subjects with binaurally normal hearing (thresholds $\leq$ 20 dB HL) were included in data analysis. However, two subjects had small hearing losses in one ear only ($<$5 dB) and one subject had a middle ear infection in one ear. Previous unpublished research in our laboratory (and the fact that these data did not manifest as outliers), has found that isolated unilateral cases such as these do not affect end results and therefore, data from these subjects were included in analysis.

## Speech-in-Noise (SiN) discrimination task

The SiN discrimination task consisted of subjects being asked to identify sentences presented in a background of multi-talker babble noise (details below). This task was administered from an HP Omnibook 4150 computer, using a program developed in-house to set noise and sentence level, to control presentation of sentences and noise, and to record, display and store results. The sentences and noise were streamed from the PC to Sennheiser HD353 headphones binaurally. Calibration of the sound stimuli was performed by coupling the headphones to a Brüel and Kjær Artificial Ear Type 4152

Table 1 **Demographic characteristics.** Demographic characteristics of students for Years 1 & 2 of MBBS undergraduate degree.

**MBBS Cohorts 2008-2010**

| Total | N |
| --- | --- |
|   Year 1 | 103 |
|   Year 2 | 54 |
| % Local:International | |
|   Year 1 | 63:37 |
|   Year 2 | 59:41 |
| % Gender | |
|   Males | 46 |
|   Females | 54 |
| Age of Acquisition of English | |
|   <5 years old | 88 |
|   >5 years old | 15 |
|   Range | 1–12 years |
| Age (years) | |
|   Mean (SD) | 19.94 (1.19) |
|   Range | 18–24 |

containing a Brüel and Kjær 1-inch Condenser Microphone Type 4145. The microphone output was connected to a Brüel and Kjær Precision Sound Level Meter Type 2203 on which sound pressure levels (SPLs) were read off (using the A-weighted scale on a slow time setting). The sentence level was standardized using a reference 1 kHz signal, with average RMS level set to the same value as for the sentences and stored on the computer as a .WAV file. Calibration of the background masking noise was done by playing the noise out of the headphones and again using the slow time settings to measure output level.

## Test sentences

Test sentences came from a standard battery of clinically-used sentences (*Bench, Kowal & Bamford, 1979*) adapted for Australian use (the BKB(A) list of sentences). The BKB list contains 192 sentences, each of 4–6 words of no more than two syllables. They are short, simple words and phrases imitating everyday speech and do not include questions or explanations open to interpretation. Also, these sentences contain words that have been shown to be very familiar to non-English speakers (*Brouwer et al., 2012*). Each sentence consists of three keywords critical for comprehension of that sentence. The sentences are pre-recorded in a female voice with an Australian accent in a neutral tone and stored as .WAV files on the computer.

Sixty sentences with similar speech reception thresholds (SRTs: the signal-to-noise ratio (SNR) at which 50% of the subjects could correctly detect the sentence in background noise) were selected for use in this study. Selection and validation of these sentences have

been detailed previously (*Burns & Rajan, 2008*; *Cainer, James & Rajan, 2008*; *Rajan & Cainer, 2008*). The sentences were randomly allocated to one of three lists classed as 'Low', 'Moderate' or 'High' to denote the level of the masking noise in which they were presented; sentence level was always set to 80dBA.

## Masking noise

The masking noise was 'babble noise' (BN), created as described previously (*Burns & Rajan, 2008*; *Cainer, James & Rajan, 2008*; *Rajan & Cainer, 2008*) to give the illusion of eight voices speaking at once, known as the 'cocktail party' effect, digitized and stored as .WAV files. Sentences were presented to subjects in a background of one of three noise levels: 1) Low noise level at 78dBA (SNR of + 2 dB); 2) Moderate noise level at 81dBA (SNR of −1 dB); and 3) High noise level at 84dBA (SNR of −4 dB). The noise was played continuously throughout each test list and was turned off at the end of each list until just before the start of the next list.

## General procedures

For the SiN discrimination task each subject was instructed that they would be presented with three lists of sentences in noise, in succession. Each list would consist of 20 different sentences in a fixed background noise level of low, moderate or high. The order of lists, i.e., test SNRs was randomised between subjects except that the high noise level list was never presented first to ensure subjects did not start with the most difficult condition. The subject was asked to repeat each sentence after it was played to the best of their ability, or to indicate if they were unable to identify it at all, with no time limit imposed on giving the response. The experimenter would score the response and then play out the next sentence. After all 20 sentences in a list had been played, this procedure would be repeated twice more, with a different list of sentences and a different noise level, until all three lists had been tested.

Upon confirmation that the subject understood the instructions and was ready to commence, the masking noise appropriate for the first test list was switched on and played by itself for 5 s before the first sentence was played. Each sentence was scored as correct only if all three keywords were identified correctly and in correct order. Once the experimenter had scored the response, the next sentence was automatically played 1.5 s later, and the test continued until all 20 sentences had been presented. Subjects were given a short break between lists. The order of presentation of sentences in each list was randomised by the software so it was unique for each subject. Scoring of performance in each list consisted of recording the percentage of sentences they were able to recall in each list.

## Indexing performance in the SiN task: calculating the SNR$_{50}$

For data analysis, the first step was to calculate the percentage of sentences identified correctly by a subject for each list. This was done using only the middle ten sentences for each noise level for the following reasons: The first five sentences were discarded as training

sentences as in our previous studies (*Burns & Rajan, 2008*; *Cainer, James & Rajan, 2008*; *Rajan & Cainer, 2008*), and the last five were discarded as some subjects showed signs of fatigue or loss of concentration.

Then data from each subject were fitted with a linear function using regression analysis and from the regression equation the midpoint of the function – the SNR at which 50% of the sentences would be detected correctly ($SNR_{50}$) was determined. These $SNR_{50}$ data represented the measure derived from the SiN task as a measure of verbal working memory. We also calculated $SNR_{50}$ using only the last 10 sentences of each list and found generally similar $SNR_{50}$ effects. We therefore chose to use the middle 10 sentences as least likely to be affected by either training effects or loss of concentration.

## Academic assessment

As well as the SiN test and questionnaire, the students' academic marks were also collected from the standard academic assessments faculty databases for data analysis. This included the first and second year data for the 2008 & 2009 cohorts, but only the first year data was collated for the 2010 cohort due to time limitations. Therefore analysis for the first year results were performed using the 103 students mentioned earlier; for the second year, analysis could be performed only on 54 (from the 103) students who had completed both years of study, i.e. students from the 2008-2009 cohorts only.

Course assessments varied from year to year, however all students' marks consisted of a combination of written examinations, individual coursework and objective structured clinical examination (OSCE) simulations. For data analysis nomenclature, these assessments were termed 'End-of-Year Totals' (Year 1 or Year 2); 'Coursework', comprising of essays, oral presentations and portfolios; 'Examinations', comprising of Multiple Choice and Short Answer Questions; and 'OSCEs' whereby the students undergo simulated clinical/patient scenarios at various timed stations whilst being assessed. The OSCEs were further subdivided into two categories according to the skills that were being evaluated: those in which the emphasis was primarily on technical skills ('OSCE Technical', e.g., injecting techniques or taking vital signs) or those in which the emphasis was primarily on communication skills ('OSCE Communications', e.g., taking a patient's history or providing an explanation to a simulated patient).

## Statistical analyses

Statistical analyses were performed using SPSS v19.0.0 (SPSS Statistics Inc.) for Windows. All statistical tests were parametric, and data were checked for normality of distribution and variation. Pearson's correlation was conducted to investigate the relationship between items from the questionnaire, Perceived Stress Scale, Index of Learning Style (the visual/verbal component only was analysed as the other components are not pertinent to this particular study) and Signal to Noise Ratio ($SNR_{50}$). Standard multiple regression was carried out to assess the relationship between language proficiency and verbal working memory ($SNR_{50}$) and hierarchical multiple regression was used to test the ability of three measures ($SNR_{50}$, age of acquisition of English and English proficiency) to predict academic performance. Student's t-tests were also used when comparing independent groups.

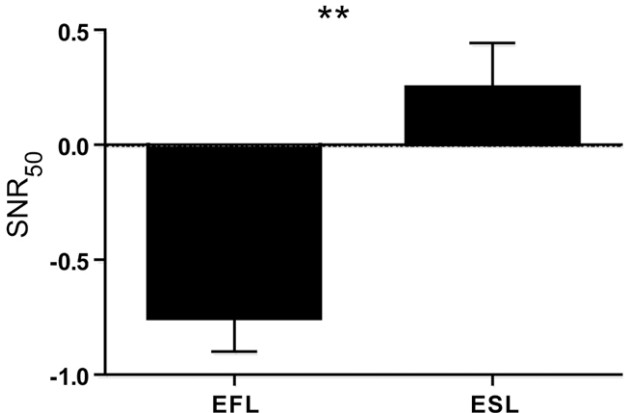

**Figure 1  SNR-50 scores for EFL vs. ESL MBBS students.** Difference in $SNR_{50}$ scores between EFL and ESL students. $SNR_{50}$ mean scores ($\pm SEM$) for students with English as first or second language. EFL: English as First Language $N = 47$. ESL: English as Second Language $N = 31$. Bilingual students were excluded $N = 25$. **$p < 0.001$.

## RESULTS

### Speech in noise performance and relationship to English proficiency

We used the SiN task to assess the presence of vWM deficits in L2 in our medical student population. In comparing across groups, students who had learnt English as a first language, had significantly smaller $SNR_{50}$ values than the students who had learnt English as a second language (*Student's-t*$(76) = -4.208, p < 0.001$) as seen in Fig. 1. Twenty-five students were not included in this analysis, as they had learnt English and another language concurrently (true bilingual) and thus did not have English as a first or second language.

These observations established that the point of subjective performance (the $SNR_{50}$) from our SiN task is a good index of verbal working memory for L2 in our medical student population.

We then used correlational analysis to assess the relationship between $SNR_{50}$ and English usage items from the questionnaire, as outlined in Table 2.

Significant negative correlations were observed between seven of the nine variables on the questionnaire and the $SNR_{50}$. The remaining two variables, Perceived Stress Scale (PSS) and the Age of Acquisition of English (AoAoE), were significantly positively correlated with the $SNR_{50}$, indicating that those with a higher $SNR_{50}$ ratio (poorer capacity to discriminate simple English sentences from noise) had learnt English later in life, i.e. more likely the international medical students, and had higher levels of stress (as noted in the current literature). Local students exhibited significantly lower $SNR_{50}$ scores than the international medical undergraduates ($t(101) = 6.23, p < 0.001$), as well as being significantly younger when they first learnt English ($t(101) = 3.33, p = 0.001$).

No significant correlation was detected between the $SNR_{50}$ and the students' Visual/Verbal Learning Style ($r = -0.023$), suggesting that the possible cultural variability in this factor was not a substantial confound in our findings.

**Table 2  Correlations of questionnaire parameters.** Descriptive statistics and Correlations table of SNR$_{50}$ and items from the questionnaire used in this study.

| | Mean (SD) | I prefer to speak English... | In the last month, how often did you speak English at home? | Perceived English proficiency | Self-rate of musical skills | Perceived Stress Scale | Visual/Verbal Learning Style Score | Age when first began playing music | Age when first learnt English | SNR$_{50}$ score |
|---|---|---|---|---|---|---|---|---|---|---|
| When I was growing up my Mother spoke English at home... | 3.78 (1.41) | .485** | .677** | **.366**** | .150 | −.236* | −.158 | −.177 | −.676** | −.465** |
| I prefer to speak English... | 4.55 (0.75) | – | .611** | .575** | .093 | −.287* | −.226* | −.238* | −.528** | −.276* |
| In the last month, how often did you speak English at home? | 4.42 (1.06) | | – | **.554**** | .142 | −.313* | −.120 | −.203* | −.654** | −.409** |
| Perceived English proficiency | 4.54 (0.78) | | | – | .279* | −.398** | −.235* | −.341** | −.483** | −.471** |
| Self-rate of musical skills | 1.89 (0.95) | | | | – | −.174 | −.071 | −.562** | −.222* | −.274* |
| Perceived Stress Scale | 13.78 (5.53) | | | | | – | −.072 | .003 | .268* | .314* |
| Visual/Verbal Learning Style Score | 4.18 (4.52) | | | | | | – | .131 | .133 | −.023 |
| Age when first began playing music | 10.49 (5.85) | | | | | | | – | .371** | .188 |
| Age when first learnt English | 2.87 (2.37) | | | | | | | | – | .394** |
| SNR$_{50}$ score | −0.30 (1.12) | | | | | | | | | – |

**Notes.**

Bolded figures are all significant at
* $p < 0.05$ or
** $p < 0.001$.

On the basis of these observations, we then conducted multiple regression analyses using the five items significantly correlated to $SNR_{50}$ that pertained to English proficiency and/or usage. These variables were: Age of Acquisition of English (AoAoE); Perceived English Proficiency (PEP); how often their mother (primary caregiver) spoke English when the student was growing up (MSE); the students' own preference for speaking English (PSE); and how often the student spoke English in the last month (ESLM). All variables were entered simultaneously using the Enter method.

The results showed that the variance in $SNR_{50}$ was significantly predicted by this model of L2 proficiency ($F(5, 93) = 9.37, p < 0.001, r^2 = 0.335$), with the five variables altogether explaining 33.5% of the total variance in $SNR_{50}$. There were two variables that significantly contributed to this overall variance. The first, Perceived English Proficiency (PEP), had the highest beta coefficient of $-0.409$ ($p < 0.001$) and accounted for 9.8% of the variance. The other variable was MSE with a beta coefficient of $-0.366$ ($p = 0.005$) and a unique contribution of 5.91% to the overall 33.5% variance. The other three variables, AoAoE, PSE and ESLM, were not significant predictors of $SNR_{50}$ in this particular model with beta values of 0.020, 0.159 and $-0.019$ respectively. However, AoAoE and ESLM showed significant correlations with $SNR_{50}$. Figure 2 graphically shows the zero-order correlations and beta coefficients for the four variables that were highly correlated to $SNR_{50}$ as also shown in Table 2.

One caveat to interpretation of our results is that the five variables pertaining to English proficiency and usage (AoAoE, PEP, MSE, PSE and ESLM) are also highly significantly correlated with each other, with $r$ values $>0.5$ (Table 2). This may suggest that these variables share the same set of underlying causal elements that affect vWM for L2 and its usage, i.e. they demonstrate multicollinearity. Therefore, a principal component analysis was performed to establish if there were underlying common constructs involved across these factors. The analysis yielded one factor with an eigenvalue $>1.0$ that accounted for 65% of the variance. All variables had high loadings with a minimum of 0.725, and a reliability test yielded a Cronbach's $\alpha$ coefficient of 0.760 (considered an acceptable value of good internal consistency).

In order to include all variables in this construct, it is necessary for all variables to be of the same scale. One variable, AoAoE, however, could not be changed (reverse coded) to the same scale as the other four variables in an appropriate way that did not change its correlation values. Therefore, it could not sit in this new construct and, as it has been widely documented that language proficiency is influenced by the age at which the language is acquired, hierarchical analysis was conducted.

The new construct of the four remaining variables, i.e. PEP, MSE, PSE and ESLM, was representative of the amount of exposure and usage the students had of English and a self-rating of their English skills. It was thus an approximation of the students' overall English proficiency, renamed 'English Language Skills' (ELS) and the means were calculated for analysis and checked for multicollinearity against $SNR_{50}$. Hierarchical multiple regression analyses were used, controlling for AoAoE in the first step and $SNR_{50}$ and the new construct ELS in the second step. Analysis was performed for the End of Year

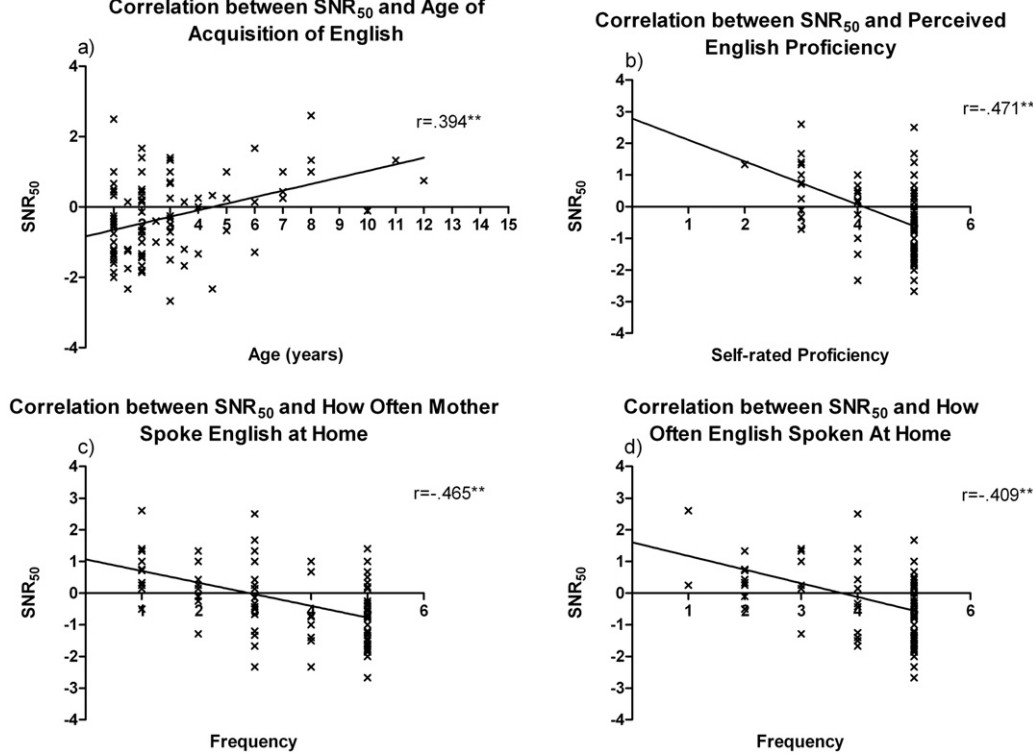

**Figure 2 Zero-order correlations between SNR-50 and English language parameters.** Significant correlations and beta values between $SNR_{50}$ and factors relating to English language skills. Figures a–c were based on answers from Likert scales ranging from 1 = poor to 5 = excellent for figure a, and from 1 = never to 5 = very often for figures b&c. $SNR_{50}$ = the Signal to Noise Ratio at which the student got 50% of the sentences correct. **$p < 0.001$, *$p < 0.05$.

Total scores, as well as for each Assessment (as described in the Methods section) for Year 1 and Year 2 of study. Results are set out in Table 3 and discussed in detail below.

These results establish that not only is $SNR_{50}$ a good index of verbal working memory for L2, but it could be employed to test if poorer L2 vWM is a strong predictor of academic performance along with language proficiency skills.

## Academic performance and relationship to English language skills

In the first year of study, the results showed that $SNR_{50}$ and ELS were not significant predictors of overall academic performance, even when AoAoE was controlled for. However, the L2 vWM index ($SNR_{50}$) did make a significant unique contribution to the OSCE Communications performance, with a beta coefficient of $-0.231$ ($p = 0.043$). This demonstrated that the smaller the $SNR_{50}$ ratio (i.e., the better the vWM for discrimination of simple English sentences from noise), then the greater the Communications score.

In contrast to this, results for the OSCE Technical skills showed significant positive correlations with the AoAoE (beta coefficient of 0.326, $p = 0.023$) and with ELS (beta coefficient of 0.329, $p = 0.030$). These correlations showed that students who had learnt English significantly later in life, but who rated their English skills more highly (international students with good English proficiency skills), performed better in the

**Peer**J

**Table 3 Analysis of results.** Hierarchical multiple regression to assess academic performance of MBBS students.

| Assessments | Mean (SD) | Predictor Variables | $R^2$ | $R^2$ Change | β | ANOVA |
|---|---|---|---|---|---|---|
| **Year 1**<br>**N=103** | | | | | | |
| **End of Year 1 Total** | 75.87 (6.17) | *Step 1:*<br>AoAoE | .003 | | −.057 | |
| | | *Step 2:*<br>AoAoE<br>SNR$_{50}$<br>ELS | .004 | .001 | −.036<br>.022<br>.041 | $F(3, 99) = .137, p = .938$ |
| **Examinations Year 1** | 72.94 (8.19) | *Step 1:*<br>AoAoE | .011 | | −.107 | |
| | | *Step 2:*<br>AoAoE<br>SNR$_{50}$<br>ELS | .014 | .002 | −.116<br>.056<br>.018 | $F(3, 99) = .462, p = .709$ |
| **Coursework Year 1** | 80.66 (8.17) | *Step 1:*<br>AoAoE | .003 | | .059 | |
| | | *Step 2:*<br>AoAoE<br>SNR$_{50}$<br>ELS | .008 | .004 | .023<br>.066<br>−.013 | $F(3, 99) = .254, p = .858$ |
| **OSCE Year 1** | 79.13 (7.83) | *Step 1:*<br>AoAoE | .001 | | .031 | |
| | | *Step 2:*<br>AoAoE<br>SNR$_{50}$<br>ELS | .034 | .033 | .176<br>−.150<br>.119 | $F(3, 99) = 1.157, p = .330$ |
| **OSCE Communications Year 1** | 78.39 (8.81) | *Step 1:*<br>AoAoE | .000 | | .003 | |
| | | *Step 2:*<br>AoAoE<br>SNR$_{50}$<br>ELS | .050 | .050 | .128<br>**−.231**[*]<br>.046 | $F(3, 99) = 1.753, p = .161$ |
| **OSCE Technical Year 1** | 81.59 (9.87) | *Step 1:*<br>AoAoE | .005 | | .073 | |
| | | *Step 2:*<br>AoAoE<br>SNR$_{50}$<br>ELS | .063 | .058 | **.326**[*]<br>−.038<br>**.329**[*] | $F(3, 99) = 2.225, p = .090$ |

Table 3 (*continued*)

| Assessments | Mean (SD) | Predictor Variables | $R^2$ | $R^2$ Change | β | ANOVA |
|---|---|---|---|---|---|---|
| **Year 2** | | | | | | |
| **N=54** | | | | | | |
| End of Year 2 Total | 74.61 (5.08) | *Step 1:* AoAoE | .012 | | −.110 | |
| | | *Step 2:* AoAoE | .026 | .014 | −.110 | $F(3, 50) = .448, p = .720$ |
| | | SNR$_{50}$ | | | .141 | |
| | | ELS | | | .064 | |
| Examinations Year 2 | 68.99 (7.49) | *Step 1:* AoAoE | .000 | | .022 | |
| | | *Step 2:* AoAoE | .012 | .012 | .058 | $F(3, 50) = .205, p = .892$ |
| | | SNR$_{50}$ | | | .131 | |
| | | ELS | | | .109 | |
| Coursework Year 2 | 80.82 (5.57) | *Step 1:* AoAoE | .026 | | −.161 | |
| | | *Step 2:* AoAoE | .043 | .018 | −.299 | $F(3, 50) = .755, p = .524$ |
| | | SNR$_{50}$ | | | .025 | |
| | | ELS | | | −.177 | |
| **OSCE Year 2** | 79.51 (6.46) | *Step 1:* AoAoE | .102 | | **−.320**[*] | |
| | | *Step 2:* AoAoE | .130 | .028 | −.209 | $F(3, 50) = 2.494, p = .071$ |
| | | SNR$_{50}$ | | | .182 | |
| | | ELS | | | .233 | |
| **OSCE Communications Year 2** | 80.45 (7.35) | *Step 1:* AoAoE | .147 | | **−.384**[*] | |
| | | *Step 2:* AoAoE | .210 | .063 | −.129 | $F(3, 50) = 4.437, p = \textbf{.008}$ |
| | | SNR$_{50}$ | | | −.068 | |
| | | ELS | | | .315 | |
| **OSCE Technical Year 2** | 77.73 (10.33) | *Step 1:* AoAoE | .006 | | −.077 | |
| | | *Step 2:* AoAoE | .110 | .104 | −.183 | $F(3, 50) = 2.05, p = .119$ |
| | | SNR$_{50}$ | | | **.346**[*] | |
| | | ELS | | | .012 | |

**Notes.**

AoAoE: Age of Acquisition of English; SNR$_{50}$: Signal-to-noise Ratio; ELS: English Language Skills.

[*] $P < 0.05$.

technical aspects of the OSCEs, despite learning the L2 at a later age. The $SNR_{50}$ was not significant, indicating that L2 vWM does not influence academic performance for this particular assessment.

Overall, after controlling for the age English was acquired, there was no clear, major predictor of academic performance in Year 1.

In Year 2, this model of vWM and ELS while controlling for AoAoE was a significant predictor of academic performance of the OSCE Communications skills ($p = 0.008$), explaining 21% of the variance of this assessment. ELS had the highest beta coefficient of 0.315 but this was not statistically significant and accounted for only 3.46% to the overall 21% variance. There was also a significant negative correlation with AoAoE on its own in Step 1 (beta coefficient $= -0.384$, $p = 0.004$), but AoAoE was no longer uniquely significant in the overall model for predicting OSCE Communication skills, indicating it has only an indirect influence on predicting performance of this academic assessment.

With regard to the OSCE Technical assessment for Year 2, the effects were incongruous with those observed in the results obtained for Year 1, with the $SNR_{50}$ now significantly correlated (beta coefficient $= 0.346$, $p = 0.038$), but AoAoE and ELS showing no correlation with academic performance. As it was the international medical students who exhibited higher $SNR_{50}$ ratios, this would indicate that these students could be performing better in this category than their local counterparts. This was confirmed by an independent samples t-test, which showed that the international medical students performed better in this assessment in Year 2 than their local peers ($t(43.73) = 3.376$, $p = 0.002$). This would suggest that the international students' L2 vWM is not impaired in this assessment in Year 2 (as in Year 1), perhaps because the recall of technical data is not as challenging on vWM capacity as conceptual and abstract comprehension (*Van Merriënboer & Sweller, 2010*). Overall, the model is not a significant predictor for this assessment and explains only 11% of the variance, with $SNR_{50}$ uniquely contributing 8.07%.

Although the model was not a significant predictor of the academic performance of the 2nd year total OSCE (i.e. not subdivided into OSCE Communications and OSCE Technical), it is worth noting that it accounts for 13% of the overall variance for this variable, which in the classroom would be regarded as a considerable proportion. T-test analysis of the Year 2 OSCE scores showed that while there was no significant difference between local and international medical students ($p = 0.113$), there was a significant difference for the AoAoE, with students who acquired English before the age of five having better overall marks for the OSCE assessment than those who acquired English later ($t(52) = 2.038$, $p = 0.047$). This is also evident in the significant negative correlation of AoAoE in Step 1, with a beta coefficient of $-0.320$ and significant $p$-value of 0.018. However, in Step 2, AoAoE was no longer significant, demonstrating that there are overlapping effects with the other variables.

To summarise, after controlling for the age at which English was first learnt, verbal working memory for English (as indexed by the $SNR_{50}$ in our speech-in-noise task) and ELS were not strong predictors of the overall End of Year Totals or for the individual Assessments, with the exception of the OSCEs. For the OSCE assessments, the contribution

made to the variance by each predictor varied for the OSCE types and was different for each year of study. The OSCE Communications was the only significant model, which in itself is a significant finding and which is discussed later.

## DISCUSSION

The relationship between verbal Working Memory and academic attainment has been well documented in L1, particularly with young learners (*Gathercole et al., 2004*). However, the role of vWM in predicting academic achievement in L2 adults, particularly medical students, has been only occasionally examined with inconsistent effects (see *Harrington & Sawyer, 1992*; *Juffs & Harrington, 2011*).

The aim of the current study was to explore if L2 vWM plays a role in academic attainment in ESL students. We indexed L2 vWM using a SiN task as a WM verbal test, as such tasks have been well documented to be a good indicator of L2 vWM and because such a task reflected, to a consistent degree, the background conditions occurring in some of the venues in which information was imparted to student doctors in their course. Linguistically, English target speech and English speech noise consist of many common properties (e.g., phonemes, syllable structures, prosodic features, etc.), which may make it more difficult for listeners, particularly non-native, to segregate target language from background noise and this may contribute to greater informational masking (e.g. *Bronkhorst, 2000*; *Brungart, 2001*; *Brungart et al., 2001*; *Lutfi, 1990*; *Rhebergen & Versfeld, 2005*; *Scott et al., 2004*; *Simpson & Cooke, 2005*; *Van Engen, 2010*; *Van Engen & Bradlow, 2007*). Background masking noise can be classed as energetic or informational; energetic masking is thought to affect speech processing at the level of the auditory periphery, whereas informational masking, e.g. babble noise, interferes with higher-order processing such as attention and cognitive load. Informational maskers have therefore been often used in working memory tasks to good effect. *Hygge, Boman & Enmarker (2003)* found that meaningful irrelevant speech noise significantly impaired recall in a text-reading memory task in 92 native high school students in Sweden.

We also examined a number of other factors known or postulated to influence L2 skills, in particular the age at which the participants first learnt English (as their L2) as this factor has previously been shown to influence English learning and proficiency (*Johnson & Newport, 1989*).

In our first analysis, we confirmed that the point of subjective performance (the $SNR_{50}$ score) in our SiN task was indeed a good index of verbal working memory for L2 in our student population, with our results showing that the EFL students had smaller $SNR_{50}$ scores than the ESL students. This meant that the EFL medical students were better able to identify simple English words in a noisy background than the ESL medical students. This was an important step as this SiN task is free of L2 proficiency concerns that have been a major criticism of previous studies that have used measures such as the Reading Span Task (*Harrington & Sawyer, 1992*; *Juffs & Harrington, 2011*) to show differences in L2 vWM and may be one explanation for the mixed findings of past studies. It is also worth noting

that *Waters & Caplan (2005)* have argued that traditional measures of WM do not relate to on-line processing of sentences, which they postulate to be due to a specialised WM system; we believe that tasks such as the SiN task are likely to be better evaluators of WM in online processing of whole connected speech.

We then used this index of vWM along with English Language Skills (ELS) as our model to predict academic attainment whilst controlling for the age that English was first acquired by the student (AoAoE).

## Different language-related factors affect different subcategories of the objective structured clinical examination assessment

In Year 1, this overall model was not a strong predictor of academic achievement, but there was a significant unique contribution of $SNR_{50}$ to the OSCE Communications score, indicating that vWM has a role in this assessment, and significant unique contributions of AoAoE and ELS to the OSCE Technical scores indicating that language fluency rather than vWM is involved in academic performance of the latter assessment. It is not surprising that the OSCE subcategories were the only assessments that showed significant correlations. This assessment type, particularly the Communications component, is one that has continually shown major performance differences between L1 and L2 medical students in many different countries and regardless of whether the L1 is English or another language (*Fernandez et al., 2007*; *Liddell & Koritsas, 2004*; *Schoonheim-Klein et al., 2007*; *Van Zanten, Boulet & McKinley, 2003*; *Wass et al., 2003*; *Woolf et al., 2007*).

We have also found similar results in a current study of a larger cohort of 872 medical students (Mann, et al., unpublished data), in which we did not measure L2 vWM or proficiency as in the present study. Our findings in this study showed that in the first year of the course, international medical students performed academically worse than their local peers in the OSCE assessment only, and not the Examinations or Coursework assessments. There were similar findings in the second year of the course; however, some groups did perform worse in all assessments including the OSCEs.

The above findings of the OSCE subcategories suggest that specifically, the memorising and automated recalling of technical information may not be as challenging to vWM as the complex task of trying to express conceptual and abstract themes (i.e. higher-order cognitive processing) by the ESL students as posited by *Van Merriënboer & Sweller (2010)*. Similarly, *Tyler (2001)* suggests that the knowledge and familiarity of a topic will determine how well a non-native speaker will perform. Therefore, factual information that is rote-learnt, such as the OSCE Technical, will be equally easy to recall for both non-native experienced and inexperienced student doctors than unfamiliar abstract or conceptual topics, such as needed in the OSCE Communication tasks, which require good verbal working memory for the L2.

Although the impairment of communication skills is more apparent in the 2nd year of study, it is important to note that we collated second year data only for the 2008 & 2009 cohorts and not for the 2010 cohort. The dynamics for the years may not be the same

and each year should be examined on its' own basis. Notwithstanding, this model again predicted academic performance in the OSCE Communication assessment, suggesting both vWM and language deficits in the ESL students affect this assessment subcategory in the second year. Similarly, whilst the OSCE Technical model that was found to apply in 1st year was not overall predictive of academic achievement, there was a significant correlation of vWM for this assessment subtype in 2nd year. Together, both OSCE subcategories point to L2 vWM impairments in these 2nd year students. This may be due to the 2nd year curriculum being more difficult than basic first year outlines, and therefore the greater demands on English language skills consequently resulting in poorer performance by the ESL. This is quite possible as *Collier (1992)* has stated that growth curves on normalized tests tend to flatten as students' progress in age and grade level and as the school load becomes academically more complex.

Overall, our model of L2 vWM and English Language Skills was a strong predictor of academic attainment (controlling for the age English was first learnt) for the OSCE Communications assessment subcategory. The fact that the Communications assessment was the only significant model is in itself significant, as although the international students have proven English proficiency (via IELTS or TOEFL), these medical students still perform academically worse than their local counterparts in this assessment, even whilst achieving higher scores for the other subjects.

Similar to the fact that we found no effects of L2 vWM on other components of assessments, in a study using L1 participants, *Kidd, Watson & Gygi (2007)* found only a weak correlation between SAT scores and auditory abilities using SiN tasks. Using a broad WM test battery, *Krumm, Ziegler & Buehner (2008)* also found only small indirect measures of WM as a predictor of academic performance. In contrast, *Tolar, Lederberg & Fletcher (2009)* found that WM strongly related to an adult's mathematical performance, but not when other cognitive factors where controlled for.

Verbal WM is not the only factor poorer for an L2 learner. *McDonald (2006)* reported that late English language learners had, in addition to poorer WM, poorer English decoding ability and lower speed of processing in English. *Takano & Noda (1993)* posited this slower speed of L2 processing as a temporary decline in thinking ability because the demanding processing load interfered strongly with the L2 subject's thinking, beyond the normal foreign language processing difficulties experienced by non-native speakers. *Takano & Noda (1995)* demonstrated that this "foreign language effect" was greater the more the foreign language was dissimilar to the native language, with greater performance differences between, for instance, Japanese and English than German and English, which share similar language roots.

It is important to note that only 51–75% of variance in academic attainment is explained by general cognitive abilities (of which processing speed and WM are two cognitive processes) (*Rohde & Thompson, 2007*). It is not surprising then that correlations among working memory (or vWM) measures, e.g. reading span, generally tend to be moderate (*Tolar, Lederberg & Fletcher, 2009*) as seen in the aforementioned studies and the results of this report.

## Limitations of the study

We have discussed our findings in relation to verbal WM as the SiN task is a verbal/auditory task and, therefore, a measure of the phonological loop of WM. We did not employ visual memory tasks, e.g. written examinations, and further research into how the mode of presentation could affect outcomes is required.

Further, we had categorised our AoAoE group as having acquired English either before or after the age of 5 years old according to extant literature. In our sample, the age range was 1–12 years, meaning that the majority of subjects in our sample learnt English pre-puberty. Most studies find greater discrepancies with L2 learners who have learnt English post-puberty (~14 years old e.g. *Mayo, Florentine & Buus, 1997*). Therefore, our results may underestimate the true effect of L2 age of acquisition on advanced learning.

## Conclusions and implications for future pedagogical design of MBBS courses

In summary, our study contributes to the growing research examining why non-native medical undergraduates generally perform academically worse than their native speaker counterparts despite having good L2 proficiency skills. The implications are that in a prestigious course such as the MBBS degree, where all students have proven high academic abilities, motivation and expectations prior to commencement, small differences at the early stages could have disproportionate impacts on the medical careers of L2 students, for example, in selection for highly competitive specialist training positions or fellowships. The knowledge from this study, therefore, could be used in the training of medical students from diverse backgrounds, for instance, by introducing compulsory language immersion programs prior to commencement of the formal course. An immersion program is typically 3–6 months and forces the student to speak and think in the host country's language in order to understand the language and the culture. Even for students who have apparently high levels of English proficiency (as gauged for our medical students by the stringent IELTS/TOEFL tests and face-to-face interviews) such immersion programs may prove to improve vWM in the language of instruction simply through more extensive use. This could be either general language immersion, or may be better if targeted to the specific clinical and health sciences language that medical students will encounter on commencement of the course. Further, advanced technology could be installed in areas of high noise conditions, e.g. audio systems in lecture theatres, that filter out 'white noise' to give better signal enhancement and brain processing of information to students. Having this information could also help medical students' in forming appropriate study habits such as understanding what is a 'good' study environment, etc.

We note that our study highlights an area where international medical students continually fall down despite rigorous processes and comparable English proficiency. Under these circumstances, we believe that our study provides a strong basis for carrying out procedures as noted above to improve equity of access by international students to resources to improve their academic outcomes.

**Abbreviations**

| | |
|---|---|
| **AoAoE** | Age of Acquisition of English |
| **BKB(A)** | Bench Kowal & Bamford (Australia) |
| **BN** | Babble Noise |
| **dB HL** | Decibels Hearing Level |
| **EFL** | English as a First Language |
| **ELS** | English Language Skills |
| **ESL** | English as a Second Language |
| **ESLM** | English Spoken In The Last Month |
| **IELTS** | International English Language Testing System |
| **L1** | First Language |
| **L2** | Second Language |
| **LOTE** | Language Other Than English |
| **MBBS** | Bachelor of Medicine/Bachelor of Surgery |
| **MSE** | Mother Spoke English |
| **MUHREC** | Monash University Human Research Ethics Committee |
| **OSCE** | Objective Structured Clinical Examinations |
| **PEP** | Perceived English Proficiency |
| **PSE** | Prefer To Speak English |
| **PSS** | Perceived Stress Scale |
| **RMS** | Root Mean Square |
| **RST** | Reading Span Test |
| **SAT** | Scholastic Aptitude Test |
| **SiN** | Speech-in-Noise |
| **SNR** | Signal-to-Noise Ratio |
| **SNR$_{50}$** | SNR at which 50% detected correctly |
| **SPL** | Sound Pressure Level |
| **SRT** | Speech Reception Thresholds |
| **TOEFL** | Test of English as a Foreign Language |
| **vWM** | Verbal Working Memory |
| **WM** | Working Memory |

## ACKNOWLEDGEMENTS

The authors wish to thank Jennifer Lindley for her vast knowledge of the MBBS curriculum and her invaluable help in recruiting the students.

### Funding

This work was funded by an Australian Postgraduate Award to CM. The funders had no role in study design, data collection and analysis, decision to publish, or preparation of the manuscript.

## Grant Disclosures

The following grant information was disclosed by the authors:
Australian Postgraduate Award.

## Competing Interests

Benedict Canny is currently Deputy Dean, MBBS Curriculum and may have had some contact with the participants in this study. However, Benedict Canny was not involved in analysis of the raw data and so viewed overall results from de-identified data only. David Reser is an Academic Editor for PeerJ.

## Author Contributions

- Collette Mann conceived and designed the experiments, performed the experiments, analyzed the data, wrote the paper.
- Benedict J. Canny analyzed the data and wrote the paper.
- David H. Reser wrote the paper.
- Ramesh Rajan conceived and designed the experiments, analyzed the data, contributed reagents/materials/analysis tools, wrote the paper.

## Human Ethics

The following information was supplied relating to ethical approvals (i.e. approving body and any reference numbers):

All ethics for this study were approved by the MUHREC (Monash University Human Research Ethics Committee), project number: CF08/2667-2008001361.

## Supplemental Information

Supplemental information for this article can be found online at http://dx.doi.org/10.7717/peerj.22.

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
