# Peer review of "Poorer verbal working memory for a second language selectively impacts academic achievement in university medical students"

_PeerJ, doi:10.7717/peerj.22_

## Round 0.1 · original submission · Minor Revisions

· Academic Editor

Minor Revisions

Thank you for submitting this impressive study for publication in PeerJ. I feel with a few revisions it will be an excellent addition to the literature.

I apologize for the delay in providing you with feedback. I had difficulty finding reviewers who had the necessary expertise necessary and were able to review the manuscript.

The two reviewers provide a wealth of useful feedback I hope you will find helpful in revising your manuscript. There are a few what I feel to be key points that I would appreciate if you could address in a revision of the paper. Along with the four items below, please consider carefully the other feedback provided by the reviewers using your own judgment on what to implement in the revision.

1. Although your study is fairly theoretical, it would seem it would have some important implications for educational practice with second language learners. As noted by both reviewers, the paper would be significantly enhanced if the discussion provided some specific suggestions on how what has been learned from your study could be applied in educational settings or in better preparing second language learners for entering these settings.

2. While the research questions and hypotheses are discussed 133-155 as noted by the second reviewer, they are not stated as clearly as they could be. Just as suggestion, along with discussing them it might be helpful to state them specifically 1, 2,3 etc.

3. As noted by the second reviewer, it is unclear how the 113 subjects for the second part of the study were selected. Please clarify how they were selected from the 582 students who participated in the first phase of the study.

4. Please check the document carefully for typographical and grammatical errors.

·

Basic reporting

The purpose of the current study is to explore whether students’ verbal working memory influences their academic achievement. The theoretical framework for this study focused on the challenges faced by students in an environment where they are learning in a second language. The authors make a compelling case for the study.

Experimental design

The authors designed an elaborate study using both self-report and observed measures. The design is novel and interesting with a comprehensive statistical analysis plan that addresses multiple confounding variables. This human subjects approved study has been rigorously conducted.

Validity of the findings

The findings are well justified. The authors provide interesting data displays that contribute to overall understanding of the results explained within the text. Study limitations are adequately addressed. Conclusions are sound, clearly reported and are connected to the original question.

Additional comments

Overall this is an excellently conducted research study. However, the educational significance is missing. What are medical educators supposed to do with these results? Nearly all, if not all, medical schools have L2 learners and would benefit from the authors’ recommendations for supporting students’ learning. Educational research must support practice.

Needed revisions:
There are numerous acronyms to keep in one’s mind while reading the paper. Is it possible to include an acronym table that can be referred to easily?

Abstract: Please explain what is meant by tertiary medical students. The readership is international and many may not be familiar with this term.

Typos:
Line 295: The word data implies plural; thus, the verb should match: was should be were.

Line 398: Cronbach is spelled incorrectly.

Reviewer 2 ·

Basic reporting

In the abstract, information like independent variable(s), dependent variable(s), expected results and actual results all should be included. However, the authors didn’t include briefly stated results in the abstract. In addition, as an expected factor which may influence the results of the study, the age of English language acquisition was left behind in the abstract too.

The paper also emphasizes the role of age of language acquisition in speech perception, mentioning several differences in the ability to process speech between monolinguals and late bilinguals. It would be better if the authors show expected and actual results about the effects of age of second language acquisition in the abstract of the paper.
In the introduction, the article should make more clarification about what the theoretical and empirical significance of the current study is and how the study can contribute to the understanding of the impact of verbal working memory on academic attainment, particularly for English as second language users.

The hypothesis/hypotheses of the study should be more clearly defined. It seemed in the abstract that the authors hypothesized poorer working memory under high noisy contexts significantly reduced the academic performance for L2. But the paper neither clearly stated the hypothesis/hypotheses or research question(s) in the discussion, nor did it explain why and how other constructs such as English proficiency (e.g., AoAoE, PEP) related to the hypothesis/hypotheses or research question(s).

Besides, the paper discussed in great details about why English proficiency would not be a confounding variable in the relationship between vWM and academic performance because all international students have to take standard tests in order to get the admission. However, this is a fair weak argument since passing stringent measures of English proficiency prior to enrolment, such as the IELTS or the TOEFL, would not guarantee that international students would performance as well as their local counterparts in academic achievements. In addition, in later examination, the paper extended to consider English proficiency as an indicator of academic achievements in the research design. It could be very confusing for readers to identify the role of English proficiency in this article.

Moreover, the evidence exhibited in the article does not sufficiently support the relationship between the “ability to process speech and to recall academic material.” Only one study (e.g., Ljung et al., 2010) from line 118 to 124 was cited for illustration. Given that this relationship is crucial for developing hypothesis/hypotheses, more relevant research from the extant literature would be better to enhance the argument.

Experimental design

1. There’s no reliability report for the Perceived Stress Scale and the Index of learning Styles Questionnaire used in the research design.

2. Stress and learning styles of the medical undergraduate students were measured as two potential factors which may affect their academic achievements. However, the authors failed to justify the reason why these two factors may or may not cause an impact. Except for two studies mentioned in line 165 to 169, there were insufficient literature reviews and explications to support this point.

3. Participants: Line 182 to 189 showed that 103 students were recruited into the complete research project. However, the authors didn’t clearly mention where they recruited the 54 participants for the year second study. Were they from the 103 participants of the first year study? Or were they from the 582 students who returned the surveys and different from the 103 students? Why there was a half loss of the sample? Would the sample loss cause any impact on the expected effect size?

4. The criterion that distinguishes “local” and “international” cannot rule out the possibility that an increasing number of students in “migrant families” who hold the permanent residency but not a L1 student or cannot speak English fluently and idiomatically. This may constitute a confound effect in the comparison between L1 and L2.

5. The authors didn’t rationalize and explain clearly the investigation of students’ academic achievements for both first and second year of the course. It is not clarified how they collected the second year’s data.

6. In terms of the limitations of the study, the within-subject experimental design may result in participants’ fatigue particularly when each participant in the study was required to listen and verbally repeat 60 sentences in total under three noise backgrounds (high, moderate and low). The author explained that high noise group would not be the first condition in order to putting the participants in a difficult situation at the very beginning. However, there is still a possibility that the participants who should have performed well scored low in the last group because of fatigue.

7. The use of everyday clinically-used sentences may pose a threat to ecological validity. Given that students are taught academic contents, which are different semantically with everyday language (e.g., academic lecture use more long sentences and professional vocabulary), in undergraduate medical courses, they may not be able to fully emerge in experimental settings.

Validity of the findings

1. From line 367 to 374, when conducting the multiple regressions, the authors used five items significantly correlated to SNR50 that pertained to English proficiency and/or usage, such as AoAoE, PEP, MSE. However, there were insufficient justifications and supportive literature review to explain why the authors did this subdividing and how did these five items pertained to English proficiency and usage.
2. In line 395, the authors did the principle component analysis for the five items pertaining to English proficiency and usage to rule out the threat of multicollinearity. What’s the VIF and tolerance for the new scale renamed “English Language Skills” (ELS)? Have the multicollinearity problem been solved successfully?
3. The authors indicated that the new ELS was an approximation of the students’ overall English proficiency. How did they draw out this conclusion? Is there any empirical evidence or related literature to support this point?
4. In the discussion section, the authors classified the background noise into 2 groups: energetic and informational, which a) did not have literature to support the categorization; b) were ambiguous in the way that how the categorization fit into the hypothesis and the design of the experiment. According to the article, the author mentioned the use of headphone to play noise in line 232 and “babble noise” in line 219, but it is not clear what category do these belong to.

Additional comments

The article includes quite a few grammatical and stylistic mistakes. For example, when listing previous literatures, “et al.” has been used as “et al” without the period in a few places. In line 489, it is not necessary to list all three authors and put on “et al.” at the end. When mentioning “local participants” and “international participants”, the capitalization of the letter "L" in the word "local" should be consistent in the paper. For example, in line 137, "L" is not capitalized, while it is in line 457. The letter "I" in the word "international" is another case. In line 516, it is better to explain what are “PSE; the SNR50” used for (are they examples or illustrations). Also, like in the line of 235, the first line of the paragraph should be indented. From the line of 479 to 485, the format of the text needs to be fixed.

---

## Round 0.2 · accepted · Accept

· Academic Editor

Accept

Thank you for addressing the feedback from the reviewers. I look forward to seeing your manuscript in print.